# The Oncogenic Signaling Disruptor, NDRG1: Molecular and Cellular Mechanisms of Activity

**DOI:** 10.3390/cells10092382

**Published:** 2021-09-10

**Authors:** Jason Chekmarev, Mahan Gholam Azad, Des R. Richardson

**Affiliations:** 1Centre for Cancer Cell Biology and Drug Discovery, Griffith Institute for Drug Discovery, Griffith University, Nathan, Brisbane, QLD 4111, Australia; jason.chekmarev@griffithuni.edu.au (J.C.); m.gholamazad@griffith.edu.au (M.G.A.); 2Department of Pathology and Biological Responses, Nagoya University Graduate School of Medicine, Nagoya 466-8550, Japan

**Keywords:** pancreatic cancer, NDRG1, thiosemicarbazone, c-Cbl

## Abstract

NDRG1 is an oncogenic signaling disruptor that plays a key role in multiple cancers, including aggressive pancreatic tumors. Recent studies have indicated a role for NDRG1 in the inhibition of multiple tyrosine kinases, including EGFR, c-Met, HER2 and HER3, etc. The mechanism of activity of NDRG1 remains unclear, but to impart some of its functions, NDRG1 binds directly to key effector molecules that play roles in tumor suppression, e.g., MIG6. More recent studies indicate that NDRG1s-inducing drugs, such as novel di-2-pyridylketone thiosemicarbazones, not only inhibit tumor growth and metastasis but also fibrous desmoplasia, which leads to chemotherapeutic resistance. The Casitas B-lineage lymphoma (c-Cbl) protein may be regulated by NDRG1, and is a crucial E3 ligase that regulates various protein tyrosine and receptor tyrosine kinases, primarily via ubiquitination. The c-Cbl protein can act as a tumor suppressor by promoting the degradation of receptor tyrosine kinases. In contrast, c-Cbl can also promote tumor development by acting as a docking protein to mediate the oncogenic c-Met/Crk/JNK and PI3K/AKT pathways. This review hypothesizes that NDRG1 could inhibit the oncogenic function of c-Cbl, which may be another mechanism of its tumor-suppressive effects.

## 1. Introduction

### 1.1. Pancreatic Cancer, One of the Most Lethal Cancers

Pancreatic cancer (PC)—or pancreatic adenocarcinoma—is often a fatal condition, killing 91–98% of those diagnosed with the disease within five years of presentation, ranking it firmly amongst the cancers with the lowest survival rate [1]. These statistics arise because most cases of PC present late, with either a late-stage local tumor or metastatic spread to other sites of the body [1]. Unfortunately, PC is rarely detected early because (1) patients often remain asymptomatic until the advanced stages of the disease, and (2) there are limited early diagnostic tools [2].

Additionally, PC is very resistant to anti-tumor therapies due to the cancer acquiring robust cytoprotective mechanisms, including desmoplasia, where a fibrous “capsule” forms around the tumor [3]. Desmoplasia is facilitated by bi-directional crosstalk between the pancreatic stellate and tumor cells, which protects the cancer cells from chemotherapy and contributes to the low survival rate of PC patients [2,3,4]. Despite the advancements in surgical and medical treatment, there has been little improvement in the survival rates of PC patients, with therapeutics only extending survival for several months or less [2]. Furthermore, as the incidence of PC increases in the developed world [1], this has created a critical need for novel and more effective diagnostic tools and treatments. Further research is required to develop early diagnostic tools (e.g., biomarkers) and better chemotherapeutics to improve survival.

The etiology of PC is poorly understood, and may involve multiple environmental and genetic factors. However, some well-established associated risk factors in its development include smoking, alcohol consumption, chronic pancreatitis, obesity, dietary factors, and infection with *Helicobacter pylori* [1]. The increasing incidence of PC in the developed world is likely to be attributable to increased modifiable risk factors, especially alcohol consumption and obesity. The non-modifiable risk factors include age, sex, ethnicity, blood group, gut microbiota, diabetes, family history and genetic susceptibility [1]. Further studies are needed to uncover any additional or emerging risk factors, and to better understand their involvement in PC development, as some have a weak correlation and need to be validated [1].

Most PC develops when the normal epithelial lining of the pancreatic ducts undergoes a series of step-wise mutations and develops into one of three main precursor lesions: pancreatic intraepithelial neoplasia (PanIN), intraductal papillary mucinous neoplasms (IPMN), or mucinous cystic neoplasms (MCN) [1]. PanINs are microscopic, non-invasive lesions that occur in the small pancreatic ducts, while IPMNs are lesions that arise in the main pancreatic ducts or their side branches [1]. Through further harmful molecular alterations, both PanIN and IPMN can develop into malignant pancreatic ductal adenocarcinoma (PDAC) [2]. The PDAC subtype makes up most exocrine tumors and constitutes >90% of all pancreatic malignancies [2]. Mucinous tumors derived from MCN lesions are the second most prevalent exocrine subtype of PC, constituting less than 10% of cases. These tumors also originate from the pancreatic ductal epithelium, but secrete mucin [2]. Other less-common subtypes of PC include pancreatic neuroendocrine tumors (pNET), constituting <5% of all PCs [2]. They are typically derived from pancreatic islet cells, and may secrete high levels of one or more pancreatic hormones such as insulin, glucagon, somatostatin, pancreatic peptide, or vasoactive intestinal peptide, depending on the cell of origin [2]. However, a large proportion of pNET are non-functional, and therefore are asymptomatic and not always readily diagnosable [5]. Another PC subtype is that arising from acinar cells, but this is rare [2].

### 1.2. Molecular Pathways Affected during PC

The defining genetic characteristic of PDAC, the most common form of PC, are activating *KRAS* mutations that occur in approximately 92% of PDAC cases [2]. Under normal physiological conditions, growth factors, hormones, or chemokines stimulate their respective receptor tyrosine kinases (RTKs) by binding to their extracellular domains [6]. This binding of the ligands leads to RTK dimerization and the autophosphorylation of their cytoplasmic domains, creating binding sites for guanine nucleotide exchange factors (GEFs), such as Son of Sevenless-1 (SOS1) [6]. With the aid of the GRB2 adaptor protein, SOS1 binds to the RTK and interacts with RAS to catalyze the exchange of GDP for GTP, forming the active RAS-GTP molecule [2,6]. This process initiates the RAS/RAF/MEK/ERK pathway, where active RAS then proceeds to bind and activate RAF [7,8]. Active RAF subsequently phosphorylates MEK, which in turn phosphorylates ERK. Then ERK phosphorylates and activates the downstream transcription factors responsible for cell-cycle progression and tumor survival [6] (Figure 1).

RAS is also involved in other oncogenic pathways, such as the PI3K/AKT pathway. Active RAS can associate directly with PI3K, and can lead to its activation and subsequent signaling [7]. Similarly to RAS, growth factors and (RTKs) also facilitate the activation of PI3K [6]. Normally, PI3K binds to the active RTK that then results in the phosphorylation of Phosphatidylinositol 4,5-bisphosphate (PIP2) to Phosphatidylinositol-3,4,5-trisphosphate (PIP3) [6,9]. Subsequently, PIP3 activates AKT via phosphorylation to form pAKT, which then activates the downstream targets involved in cell growth (e.g., mTORC1), survival (e.g., MDM2) and metastasis (e.g., NF-κB) [6]. An important negative regulator of the PI3K/AKT pathway is the phosphatase and tensin homolog deleted on chromosome 10 (PTEN), which dephosphorylates PIP3 to PIP2 [6,9].

In PDAC, one of the RAS family members, KRAS, becomes constitutively active, causing the PI3K/AKT and RAS/MEK/ERK pathways also to become constitutively active, contributing to the uncontrolled proliferation and survival of PC cells observed in PDAC [2,9] (Figure 1). However, studies suggest that the activating *KRAS* mutations alone is not sufficient to enable progress into malignant PDAC [2]. In fact, *KRAS*-activating mutations in pancreatic mucosal cells readily develop into PanIN precursor lesions [2]. However, in order to progress into PDAC, there must also be an additional inactivation of key tumor suppressor genes, the most common being *cyclin-dependent kinase inhibitor 2A (CDKN2A)*, *p53*, and *SMAD4* [2].

## 2. Other Molecular Pathways Affected in PC

Many other signaling pathways can contribute to the development of PC. These pathways can become deregulated due to abnormal autocrine or paracrine signaling and somatic mutations that amplify gene expression or cause constitutively active signaling [7]. The NF-κB, TGF-β/SMAD and ERBB receptor family signaling pathways have been demonstrated to promote PC growth and metastasis [10,11,12,13].

### 2.1. NF-κB Pathway

The nuclear factor kappa B (NF-κB) pathway has been implicated to play a role in the pathogenesis of many tumors, including PC [10,14,15]. In chronic inflammatory conditions such as pancreatitis, NF-κB activity is elevated and promotes a pro-tumorigenic effect [10,14]. The NF-κB pathway is stimulated by many factors, including cytokines during inflammation, growth factors, bacterial and viral products, radiation, reactive oxygen species (ROS), and DNA damage [14]. NF-κB signaling is divided into the canonical and alternate pathways (Figure 2). First, considering the canonical pathway, NF-κB dimers containing p65 (RelA) or c-Rel, together with p50 (NF-κB1), are maintained in an inactive state in the cytoplasm by the inhibitor of NF-κBα (IκBα) [16,17] (Figure 2). Activation by the previously mentioned stimuli leads to the activation of the IκB kinase (IKK) complex (consisting of IKKα, IKKβ and IKKγ subunits), which then phosphorylates IκBα, marking it for degradation [16,17]. The degradation of IκBα then allows the c-Rel/RelA-p50 dimer to translocate to the nucleus, where it transcribes oncogenic targets [14,16] (Figure 2). The alternate pathway involves the NF-κB dimer of RelB binding to p100 (NF-κB2), in which the IKKα homodimer phosphorylates p100, whicht then undergoes proteolysis to generate p52 [16,17] (Figure 2). The active RelB-p52 dimer then translocates to the nucleus [16,17].

Once in the nucleus, the NF-κB dimer undergoes a series of post-translational modifications and induces the expression of various anti-apoptotic factors involved in cell cycle progression, such as c-myc, various cyclins, and angiogenic factors [14,16]. NF-κB also regulates metastasis via transcription factors such as TWIST and SNAIL, which induce the epithelial–mesenchymal transition (EMT) [14]. Furthermore, TWIST and SNAIL up-regulate cell adhesion molecules, such as selectins and integrins, which mediate cancer cell extravasation and colonization to distant sites [14]. The expression of NF-κB also activates stress fibers important for cell migration that is critical for metastasis [14,18,19]. This multitude of effects demonstrates that NF-κB has a key role in cancer progression.

### 2.2. TGF-β/SMAD Pathway

The canonical TGF-β pathway acts as a tumor suppressor in the early stages of PC by promoting apoptosis and cell cycle arrest [12], but can then also act in a pro-oncogenic manner in advanced PC by promoting metastasis [11,20]. The same effect can be attributed to many other types of cancer [21]. In brief, the pathway is initiated with the binding of the TGF-β ligand to the TGF-β type II receptor (TβRII) [12,22] (Figure 3). This interaction with the ligand induces the TGF-β type I receptor (TβRI) to form a hetero-tetrameric complex with TβRII [12,22]. Then TβRI phosphorylates SMAD2 and SMAD3, which subsequently forms a complex with SMAD4 [12,22]. The SMAD2/3/4 complex translocates to the nucleus, recruits co-transcriptional factors, and activates the genes involved in apoptosis and cell cycle arrest [12,22] (Figure 3). For example, the SMAD2/3/4 complex promotes the transcription of the cyclin-dependent kinase (CDK) inhibitor, p21, leading to cell cycle arrest and various molecules that initiate apoptosis, such as the death-associated protein kinase [12,22].

Due to its anti-tumor effects, the TGF-β/SMAD pathway is inactivated in many cancers [22]. However, in advanced cancer, tumor cells can exploit the TGF-β/SMAD pathway to promote cancer progression through the induction of epithelial-mesenchymal transition (EMT) [22], a major step in cancer progression to a more deadly metastatic state. In advanced tumors, the SMAD2/3/4 complex promotes the expression of SNAIL, SLUG, TWIST and ZEB, which are mesenchymal markers that are transcription factors known to be involved in the EMT [12,22] (Figure 3).

### 2.3. The ERBB Receptor Family

The ERBB family of receptors consists of the epidermal growth factor receptor (EGFR), HER2, HER3 and HER4, with EGFR being the most extensively characterized [13,23]. These receptors are activated by many ligands, including epidermal growth factor (EGF), transforming growth factor α (TGFα), and many others that either bind to specific or multiple ERBB receptors [13]. Ligand binding facilitates these receptors to form homodimers or heterodimers with each other, which is required for their activation through the receptor trans-autophosphorylation of their cytoplasmic domains [13,23].

The phosphorylated tyrosine residues of the activated ERBB receptor are then recognized by many downstream signaling proteins, which bind the receptor and form large signaling complexes [13]. This event results in the activation of many signaling pathways, but the pathways targeted depend on the type of ERBB receptor and their dimer partners [13,23]. For example, the HER2/HER3 dimer particularly stimulates the PI3K pathway [13,23]. Some pathways activated by EGFR, HER2 and HER3 receptors include the TGF-β, WNT, RAS, c-Src, PI3K, MAPK and NF-κB pathways involved in proliferation, migration, differentiation and survival [13,23]. Aberrant ERBB receptor activation and kinase activity contribute to the progression of many cancers, including pancreatic, prostate, colon, colorectal, anal, lung, breast and oesophageal cancers, and melanoma, etc. [13]. Of note, the function of the HER4 ERBB receptor remains the least well characterized of this protein family, with its roles yet to be firmly established [13,23].

## 3. Current Treatments for PC

The only treatment that offers a potential cure for PC is surgical resection, an example being the Whipple procedure [1,24]. This procedure may be performed for PC located in the head of the pancreas, and it involves removing the head of the pancreas, the duodenum, part of the stomach, the gallbladder, and part of the bile duct [24]. Then, the digestive and biliary tracts must be reconnected [24]. Unfortunately, 80–85% of PC cases are not resectable at the time of diagnosis [1], reflecting a need for better treatments and methods for early diagnosis. Although the recurrence rates are high after resection, adjuvant chemotherapy (that is treatment after surgery) has been shown to improve survival [1]. Resection followed by adjuvant chemotherapy with gemcitabine and capecitabine is the current standard treatment for resectable PC [25,26]. On the other hand, adjuvant chemoradiotherapy is not recommended, and has been shown to worsen survival rates in clinical trials [26].

Classically, neo-adjuvant chemotherapy, which is chemotherapy prior to surgery, is implemented with the FOLFIRINOX combination of drugs, which includes: (1) leucovorin (folinic acid, a folate analog), (2) 5-fluorouracil (a nucleic acid synthesis inhibitor), (3) irinotecan (a topoisomerase inhibitor), and (4) oxaliplatin (a DNA damaging agent). Alternatively, the treatment can implement the GnP combination, namely gemcitabine (a DNA synthesis inhibitor) and nab-paclitaxel (a microtubule inhibitor) [27]. These combinations are the accepted standard to treat borderline resectable PC, and have been shown to improve outcomes after resection [27]. More recently, the recommendation for chemotherapy of PDAC was revised [28]. Accordingly, the American Society of Clinical Oncology (ASCO) clinical practice guideline [28] recommends the combination chemotherapy regimen of 5-fluorouracil, oxaliplatin and irinotecan (modified FOLFIRINOX, also known as mFOLFIRINOX) as the preferred adjuvant therapy for patients with PC and an ECOG performance status of 0–1 who have undergone an R0 or R1 resection and have not received neoadjuvant chemotherapy. For patients with a poor general condition or contraindications to mFOLFIRINOX, gemcitabine monotherapy is available. In the case of intolerance to gemcitabine, 5-fluoruracil/leucovorin is available as a therapeutic alternative.

However, neo-adjuvant therapy can result in complications and delay the resection of tumors, which can continue to progress until it is too late for surgical intervention [1].

Most PC cases are diagnosed in the later stages of the disease, where metastasis has already occurred [1]. The primary treatment for advanced or metastatic PC includes the FOLFIRINOX regimen, with complementary jaundice and symptom management [1]. FOLFIRINOX therapy can have adverse effects, but demonstrates significantly improved survival compared to gemcitabine alone, the previous gold standard chemotherapeutic for PC [1,25]. Furthermore, unfortunately, PC cells have shown resistance to adjuvant, neo-adjuvant, and metastatic treatments with FOLFIRINOX or GnP [25]. The use of immunotherapy to treat PC has shown minimal success to date, with a number of immunotherapies being in development [29,30]. However, there have been some promising results in mouse models, where immune therapies together with chemotherapy have shown synergistic responses [30].

Despite the various PC treatments described above, PC mortality has remained high, at around 95% [31]. In addition, these treatments are associated with adverse effects [32,33,34,35]. Hence, less toxic and more effective therapies are desperately sought. Considering this, it is critical to investigate new classes of chemotherapeutic agents. In this review, the selective anti-neoplastic activity of innovative agents that bind tumor cell iron and copper as selective anti-tumor agents will be discussed, with an emphasis on the molecular mechanism of their activity in terms of their effects on oncogenic signaling [4,36,37].

## 4. Iron Is Crucial for Cell Survival and Proliferation

Iron is an essential nutrient involved in almost all cellular processes [36]. Numerous enzymes and proteins that play a role in oxygen transport, DNA synthesis, oxidative metabolism, energy generation and cell proliferation require iron as a co-factor in their active site to function [36,38]. These proteins include ribonucleotide reductase (RR), hemoglobin, myoglobin, and cytochrome *c*, etc. [36,39]. Iron is also important in the regulation of the proteins involved in cell cycle progression, such as cyclins, p53, retinoblastoma protein, and cyclin-dependent kinases (CDKs) [36]. Therefore, iron is crucial for cell survival and proliferation.

The higher rate of cancer cell proliferation than normal cells causes them to have higher requirements for iron than normal cells [40,41,42]. This need is reflected by cancer cells having a higher expression of the TfR1 on their membranes, allowing the greater uptake of transferrin-bound iron from the bloodstream [41,42]. The expression of the iron storage protein, ferritin, was shown to be decreased in some cancers, potentially allowing more iron to reside in the labile iron pool (LIP) and be used for cellular metabolism [43]. In contrast, some tumors demonstrate the up-regulated expression of ferritin for increased iron storage [41,42]. Furthermore, the endosomal reductase, STEAP3, has been demonstrated to be highly expressed in cancers, and is thought to reduce transferrin Fe^3+^ to Fe^2+^ in the endosome to be exported into the cytosol for cellular use [41].

The expression of the iron export protein, ferroportin1, is also decreased in cancer cells, reducing the iron export out of the cell, which is an advantage for tumor cell proliferation that requires iron [41,44]. Collectively, these alterations in the expression of the proteins involved in iron metabolism increase the iron levels in cancer cells, providing sufficient iron for RR activity that is required to catalyze the reduction of ribonucleotides into deoxyribonucleotides for DNA synthesis [41,42]. Overall, cancer cells are highly dependent on iron compared to normal cells, making them more vulnerable to iron depletion. This characteristic can be targeted and exploited by specifically designed anti-cancer agents that ligate cancer cell iron pools. Cancer cells also take up more copper—a metal ion required for angiogenesis, invasion and metastasis—than normal cells [40]. Again, this characteristic can be taken advantage of by using agents that ligate both iron and copper [36].

## 5. Cancer Therapy Using the Metal-Binding Ligands DFO and Thiosemicarbazones, Dp44mT and DpC

### 5.1. Desferrioxamine

The use of specifically designed metal-binding ligands for cancer therapy is a relatively recent therapeutic strategy. Initially, iron-binding ligands, including desferrioxamine (DFO) (Figure 4), were developed in order to treat iron overload diseases such as β-thalassemia [36,45]. However, it was later demonstrated that these agents also possessed anti-cancer activity [42,46]. For example, DFO displayed an anti-proliferative activity against neuroblastoma and leukemia in vitro, in vivo using animal models, and in human clinical trials [47,48]. DFO is a naturally occurring and canonical iron chelator, which was one of the first ligands to be implemented in anti-cancer studies [36]. DFO binds Fe^3+^ within cells or free plasma Fe^3+^, creating a complex (ferrioxamine) that is redox and metabolically inert [36]. Once DFO binds intracellular iron [49], it is then released from cells and excreted primarily in the urine, with a small component entering the bile and feces, resulting in whole-body iron depletion [50]. While DFO has been demonstrated to inhibit neuroblastoma cell lines in vitro [36,47], it is less effective in vivo due to its rapid metabolism and poor lipophilicity, which limits its ability to enter cells [36].

### 5.2. The Thiosemicarbazones, Dp44mT and DpC

The suboptimal properties of DFO as an anti-cancer agent led to the specific design and development of ligands that demonstrated more potent anti-proliferative activity, such as those of the pyridoxal isonicotinoyl hydrazone class [51,52,53,54,55]. Successive design iterations of these agents [56,57] led to the development of the di-2-pyridylketone thiosemicarbazone series of ligands, such as the lead compounds di-2-pyridylketone 4,4-dimethyl-3-thiosemicarbazone (Dp44mT) and di-2-pyridylketone 4-cyclohexyl-4-methyl-3-thiosemicarbazone (DpC; Figure 4), which demonstrate marked activity in vitro and in vivo against a range of tumors [36,58,59,60,61]. Both Dp44mT and DpC have chemical properties that make them more lipophilic than DFO, which promotes their membrane permeability and ability to bind intracellular iron [36].

The ability of these agents to bind intracellular iron results in inhibitory activity on a variety of molecular targets, including the iron-dependent enzyme, RR, that is the rate-limiting step of DNA synthesis, as well as proteins involved in proliferation and the cell cycle, e.g., cyclin D1 and cyclin-dependent kinase 2 [36] (Figure 5). Importantly, iron depletion also increased the expression of the potent metastasis suppressor, NDRG1 [62,63,64], which will be discussed in more detail in Section 6.3.

### 5.3. Thiosemicarbazones: More than Just Iron-Binding Ligands

The activity of the di-2-pyridylketone thiosemicarbazones goes beyond merely depleting tumor cell iron levels. Thiosemicarbazones also bind to and deprive cancer cells of copper [65], which is essential for angiogenesis (Figure 5), a critical process that facilitates cancer metastasis and invasion [66]. Copper is also needed for other essential enzymes, including mitochondrial cytochrome *c* oxidase, which is critical for energy generation; lysyl oxidase, which is needed for connective tissue formation; and superoxide dismutase, which protects against cytopathic superoxide [66]. Indeed, copper depletion has been shown to inhibit a range of cancers [67,68,69]. Thiosemicarbazones act by a “double punch” mechanism by binding intracellular iron and copper, and forming redox-active complexes that generate ROS [36,70]. The ability to result in ROS leads to a decrease in glutathione (GSH) and an increase in its oxidized form (GSSG) [40]. Predictably, increased GSH levels decrease the cytotoxic effects of Dp44mT, while GSH depletion increases the anti-proliferative activity of this agent [40].

An essential aspect of di-2-pyridylketone thiosemicarbazone activity is its ability to be lysosomotropic and accumulate in lysosomes via the P-glycoprotein (Pgp) drug efflux pump [71]. Classically, Pgp promotes drug resistance in many cancers by transporting drugs out of the tumor cell when this protein is expressed on the plasma membrane [71]. More recently, it has been discovered that Pgp is present not only on the plasma membrane but is also expressed on endosomal and lysosomal membranes, where it demonstrates functional activity (Figure 6) [72,73]. This intracellular distribution of Pgp results in the transport of its substrates into endosomes or lysosomes, and in the case of the thiosemicarbazones, it enables facilitated access to one of their primary targets, namely the lysosome [40].

Once it is within this organelle, the thiosemicarbazones bind to the iron and copper released from the lysosomal turnover of iron- and copper-containing proteins. The formation of these thiosemicarbazone-iron and copper complexes is critical, as they demonstrate redox activity, which results in ROS generation and lysosomal membrane permeabilization that leads to apoptosis (Figure 6) [36,40,74].

### 5.4. The Selectivity of Thiosemicarbazones

At optimal doses, the di-2-pyridylketone thiosemicarbazones demonstrate good selectivity against cancer cells relative to normal cells [59]. This selectivity can be rationalized to be due to the elevated replication rate of tumor cells, which results in a greater requirement for iron and copper for essential metabolic processes (e.g., DNA synthesis), making them more sensitive than normal cells [36]. Additionally, as mentioned above, thiosemicarbazones target lysosomes via the Pgp that is expressed in these organelles, as well as the plasma membrane (Figure 6) [71]. The expression of Pgp is observed in over half of all human cancers, including PC, which facilitates the activity of the thiosemicarbazones in a broad range of tumors [71].

Interestingly, *RAS*-mutated cancers—such as lung, colon, and pancreatic cancers—have higher autophagy and lysosomal activity, which facilitates the recycling of cellular constituents that promote tumor growth [75]. As such, under these latter conditions, thiosemicarbazones should possess potentiated efficacy. Considering the analysis above, three characteristics of cancer cells could potentiate their sensitivity to the di-2-pyridylketone thiosemicarbazones relative to normal cells, namely: **(1)** high iron and copper dependency, **(2)** the expression of the Pgp transporter, and **(3)** the increased lysosomal activity of cancer cells [76].

### 5.5. DpC: A Second-Generation Thiosemicarbazone

Despite Dp44mT’s effectiveness both in vitro and in vivo, and the fact it was the lead agent of the first generation of di-2-pyridylketone thiosemicarbazones, it was found to demonstrate several key disadvantages. These include: **(1)** the ability of Dp44mT to cause cardiac fibrosis after intensive intravenous injections at relatively high doses; **(2)** causing significant toxicity upon oral administration; and **(3)** Dp44mT caused the significant oxidation of hemoglobin and myoglobin leading to met-hemoglobin and met-myoglobin that cannot bind O_2_ [36,77]. Hence, this led to the development of the more tolerable second-generation analogs, including DpC [36], which did not induce cardiotoxicity or oxyhemoglobin oxidation [36,77], was tolerable after both oral or intravenous administration, and inhibited PC xenograft growth more effectively than Dp44mT in vivo [63]. As an important consequence of its excellent anti-tumor activity, DpC entered phase I clinical trials in 2016 [78].

## 6. N-myc Downstream-Regulated Gene 1 (NDRG1)

Considering the multiple mechanisms of action of metal-binding ligands such as DFO, Dp44mT, and DpC, a key effector of activity is mediated through their ability to up-regulate the potent metastasis suppressor, NDRG1. Due to the importance of this protein in the anti-cancer efficacy of these latter drugs, the structure and function of NDRG1 will be described in detail below.

### 6.1. Molecular Structure of NDRG1

NDRG1 is a member of the NDRG family of proteins, and has been extensively shown to have anti-oncogenic and anti-metastatic effects in various cancers, including tumors of the brain, breast, prostate, colon, rectum, esophagus and pancreas [70,74]. Interestingly, NDRG1 has also been suggested to promote tumorigenesis in kidney, liver and oesophageal cancers [70]. This pleiotropy in NDRG1 function is potentially a reflection of the heterogenous nature of the signaling between tumor cell-types. The *NDRG1* gene is located on chromosome 8 at locus 8q24.2 [18,70]. This gene encodes a 2997-bp mRNA that is translated into a mature protein with an estimated molecular weight of 43 kDa, containing 394 amino acids [18,70].

Members of the NDRG1 family all possess an NDR domain consisting of an esterase-/lipase-/thioesterase-active-site serine and an α/β hydrolase fold that covers approximately 220 amino acids of the protein, making them a member of the α/β hydrolase superfamily [70,74] (Figure 7). NDRG1 is distinct from other NDRG proteins because it contains three decapeptide tandem repeats of the residue GTRSRSHTSE and a phosphopantetheine attachment site (PPAS) in the *C*-terminus region, as well as a cap-like domain within the α/β-fold [18,70,79] (Figure 7). Serum- and glucocorticoid-induced kinase (SGK)-1 can phosphorylate NDRG1 at Thr328, Ser330, Thr346, Thr356 and Thr366, which enables glycogen synthase kinase (GSK-3β) to subsequently phosphorylate NDRG1 at Ser342, Ser352 and Ser362 [18,70]. Calmodulin kinase II (CaMK-II) and protein kinase A (PKA) and C have also been found to phosphorylate NDRG1 in mast cells, mainly at serine and threonine residues, which has been associated with degranulation and exocytosis in this cell type [70].

The phosphorylation of NDRG1 has been suggested to influence its intracellular localization [70]. In fact, phosphorylation is crucial to activate NDRG1’s multiple physiological functions, such as differentiation, the suppression of NF-κB signaling, and the expression of CXC chemokines that promote tumorigenesis [70]. Another significant structural feature of NDRG1 is the presence of a *C*-terminal domain that binds Ni^2+^ and Cu^2+^, which is thought to be involved in NDRG1 expression and detoxification [18,70].

### 6.2. Intracellular Localization of NDRG1

NDRG1 is most commonly and predominantly localized in the cytoplasm of the cell [79,80]. However, its distribution can be dependent on the cell-type where NDRG1 is expressed [70,80]. For instance, NDRG1 is primarily localized in the plasma membrane of intestinal and lactating breast epithelial cells, the nucleus of prostate epithelial cells, and the mitochondrial inner membrane of renal proximal tubule cells [70,74].

NDRG1 has no detectable nuclear localization signal, yet it has been observed in the nucleus of several cell types [81]. Furthermore, it has been speculated that the phosphorylation of NDRG1 or its association with nuclear proteins such as Hsc70 mediates its nuclear translocation [81]. Both DNA damage and hypoxia have been demonstrated to induce NDRG1 translocation from the cytoplasm to the nucleus, suggesting that NDRG1 may act as a stress response gene [18,70]. Moreover, there is strong evidence that NDRG1 is involved in the function of cell adhesion, as it associates with cell membranes, as well as desmosomes and intermediate and microfilament bundles that insert into adherens junctions [81]. In fact, studies with cells that over-express NDRG1 have demonstrated that it maintains β-catenin and E-cadherin at the membrane, which stabilizes the adherens junctions between cells, maintaining them in a non-invasive, epithelial phenotype [82].

### 6.3. How NDRG1 Is Regulated

An array of chemical and biological factors regulate NDRG1 expression (Figure 8) [18,70]. Both DNA methylation and histone deacetylation have been demonstrated to suppress NDRG1 expression [74,83]. Numerous agents are known to promote NDRG1 expression, including DNA-damaging agents, androgens, vitamins A and D3, forskolin, lysophosphatidylcholine, tunicamycin, sulfhydryl-reducing agents, ligands of the peroxisome proliferator-activated receptor, retinoid X receptor, okadaic acid, Ca^2+^, heavy metal ions, and iron-depletion [18,70,74].

Iron depletion induced by iron-binding ligands, such as DFO, Dp44mT and DpC, markedly up-regulates the expression of the potent metastasis suppressor, NDRG1 (Figure 5 and Figure 8), through HIF-1α-dependent and -independent mechanisms [62,63,64]. The up-regulation of NDRG1 by iron-binding ligands explains the ability of iron depletion to suppress a plethora of oncogenic pathways responsible for growth and metastasis [36]. This aspect will be discussed in greater detail below in later sections.

In terms of the molecular mechanism involved in the HIF-1-mediated up-regulation of NDRG1, it is notable that HIF-1 is a heterodimeric protein that consists of the HIF-1α and HIF-1β subunits, the former of which is regulated by oxygen availability [84]. At normal oxygen levels, HIF-1α subunits are hydroxylated by prolyl hydroxylase (PHD) enzymes that create a signal for the von Hippel–Lindau (VHL) protein to bind and subsequently activate ubiquitin E3 ligase, resulting in the proteasomal degradation of HIF-1α [85] (Figure 9). On the other hand, HIF-1β subunits are constitutively expressed and not subject to the modifications of the HIF-1α subunit [84]. The hydroxylase ability of PHD is oxygen-dependent [85]. Thus, when oxygen levels are low in hypoxic states, HIF-1α is not modified by PHD and allowed to accumulate to form heterodimers with HIF-1β subunits, forming the HIF-1 complex, which undergoes nuclear translocation to then bind the hypoxia response elements (HREs) in the promotors of various genes [85,86] (Figure 9). *NDRG1* has two HREs upstream of its promotor, allowing the HIF-1 complex to induce its expression [62]. PHDs also require iron as part of their active site, so that upon the depletion of the cellular iron levels, this enzyme becomes inactive, mimicking hypoxia and preventing HIF-1α degradation [85]. HIF-1α then accumulates and forms the HIF-1 complex that induces NDRG1 expression, as observed under hypoxia [85].

Hypoxia also up-regulates NDRG1 by an HIF-1α-independent mechanism. For example, the epithelial growth response-1 (Egr-1) transcription factor has been confirmed to bind to the *NDRG1* promotor and induce its transcription under hypoxia [70,74]. Additionally, during chronic states of hypoxia, such as those in tumors, Ca^2+^ is released from intracellular stores, which induces the expression of hypoxic genes—including NDRG1—via the transcription factor, AP-1 [87]. Studies have also demonstrated that the eukaryotic initiation factor 3a (eIF3a) can prioritize the translation of *NDRG1* mRNA via the formation of stress granules that occur during cellular stress produced by iron depletion or hypoxia [64]. Thus, eIF3a is another example of a HIF-1α-independent mechanism by which iron-binding ligands induce NDRG1 expression.

Other transcription factors that regulate NDRG1 expression include p53 in response to DNA damage [36], v-ets avian erythroblastosis virus E26 oncogene homolog 2 (ETS), and PTEN [70] (Figure 8). Considering PTEN, this protein was demonstrated to up-regulate NDRG1 expression in prostate and breast cancer cells [18,70,88]. Interestingly, the reverse of this effect was observed in PC, where NDRG1 was reported to increase PTEN expression [18,70]. Hence, a positive feedback loop may be established between these molecules [18,70].

In contrast to the range of positive effectors, NDRG1 expression is also suppressed via N-myc and c-myc, potentially through the methylation of the *NDRG1* promotor, as myc proteins bind CpG islands in DNA and have binding sites in the *NDRG1* promotor [89]. The up-regulation of N-myc and c-myc occur in tumors such as prostate, breast, colon and esophageal cancer [36]. Moreover, it is speculated that trypsins cleave NDRG1 in some cancers, abrogating its metastasis suppression capabilities [90] (Figure 8).

### 6.4. NDRG1 Biological Functions

NDRG1 is involved in many biological functions, and does not seem to have one distinct purpose [18,70,74]. NDRG1 is involved in embryogenesis and development, and contributes to the proper development of the urinary and reproductive organs [70,74]. Its expression correlates with certain cell cycle phases, and it can associate with the centromeres and stabilize the spindle structure, indicating a role in cell division [70,74]. The over-expression of NDRG1 results in the up-regulation of epithelial markers such as β-catenin and E-cadherin in colon and prostate cancer [70,82], while differentiation-promoting ligands such as PPAR-γ, vitamin D and isobutyl methylxanthine have been demonstrated to induce NDRG1 expression [70]. It is known that NDRG1 is involved in lipid metabolism and the myelination of neurons, as the expression of truncated isoforms of NDRG1 through mutations has been associated with demyelinating diseases, such as hereditary motor and sensory neuropathy-Lom [70,80].

The up-regulation of NDRG1 during cellular stress, DNA damage, and hypoxia supports the idea that NDRG1 is a stress response gene possessing cytoprotective functions [18,70,74]. In fact, NDRG1 can inhibit p53 expression in trophoblasts under hypoxic conditions, protecting them from apoptosis [91]. NDRG1 also has an important role in mast cell function and inflammation, as the maturation of primitive mouse bone-marrow-derived mast cells into mature connective tissue mast cells is mediated by NDRG1 [92]. Furthermore, NDRG1 facilitates the degranulation of these mast cells in response to stimuli leading to inflammatory responses [93].

## 7. NDRG1 Inhibits Metastasis

Metastasis is the spread of malignant cells from a primary site to distant sites, and is the leading cause of death by cancer, as primary tumors account for less than 10% of cancer deaths [36]. Therefore, the process of metastasis has attracted much interest in terms of cancer therapy, as understanding how to inhibit metastasis may be a crucial step in improving cancer survival rates. Numerous studies have demonstrated the ability of NDRG1 to suppress tumor growth, angiogenesis and metastasis [36,70,74,82,88,94], processes which are critical for the progression of cancer. The mechanisms behind the anti-cancer and anti-metastatic activity of NDRG1 in terms of its anti-oncogenic effector activity against the TGF-β, Wnt, NFκB, PI3K/AKT, RAS/RAF/MEK/ERK, ERBB receptors, and the Sonic Hedgehog oncogenic signaling pathway are discussed below

### 7.1. NDRG1 Inhibits the Epithelial–Mesenchymal Transition (EMT) via the TGF-β/SMAD Pathway

Metastasis is a complex process that enables the invasion and migration of cancer cells throughout the host [36,95], with the EMT being a key initial step in the metastatic cascade [36,95]. The EMT involves epithelial cells held adjacent to each other by adherens junctions transitioning into cells with a mesenchymal phenotype [95]. This process results in the down-regulation of “epithelial” proteins, such as E-cadherin and β-catenin, which compose the adherens complex, resulting in the loss of adherens junctions and cell polarity and the up-regulation of “mesenchymal” proteins such as vimentin, tenascin C, laminin β1, collagen type VI α, and various proteinases [95]. All of these factors result in cells with increased motility and invasive potential, which can initiate metastasis. Outside the context of cancer, the EMT is crucial for embryonic development, wound healing and tissue remodeling [36,95].

It is known that transforming growth factor-β (TGF-β) induces the EMT in various cell-types, including cancer cells, causing the loss of E-cadherin and β-catenin at the cell membrane [36,82]. Studies in prostate and colorectal cancer cells have demonstrated that the over-expression of NDRG1 inhibited TGF-β-induced EMT by inhibiting SMAD2 and pSMAD3 expression, which are key signaling molecules involved in the TGF-β/SMAD pathway [36,82]. This mechanism prevents the TGF-β-induced expression of transcription factors such as SNAIL, SLUG and TWIST, which repress E-cadherin expression [36,82]. With SNAIL, SLUG and TWIST no longer repressing E-cadherin due to NDRG1, this enhances the formation of the adherens junction, retaining the epithelial phenotype and suppressing motility and metastasis [36,82]. Iron-binding ligands, such as DFO and Dp44mT, inhibit the TGF-β-induced EMT via NDRG1 up-regulation, providing a potential therapeutic strategy to inhibit tumor metastasis [36,82].

### 7.2. NDRG1 Inhibits the Wnt/β-Catenin Pathway

Wnt signaling activates a number of pathways [96]; here, we will focus on the canonical Wnt/β-catenin pathway. The levels of cytoplasmic β-catenin are normally tightly regulated by its degradation, mediated via phosphorylation by the destruction complex, consisting of casein kinase 1α (CK1α), glycogen synthase kinase-3β (GSK3β adenomatous polyposis coli (APC), the scaffold protein, Axin, and other proteins [96,97,98] (Figure 10). However, upon Wnt ligand binding to the Frizzled and LRP5/6 co-receptors, the Dishevelled (DVL) protein is recruited to the receptor and inhibits the destruction complex by sequestering it to the membrane [97,99]. This process allows cytosolic β-catenin to accumulate and translocate to the nucleus to act as a co-activator with the T cell factor (TCF) family transcription factors [97,99]. The result of this activity is the transactivation of oncogenic target genes such as those encoding cyclin D1 and c-myc [97,99] (Figure 10). Indeed, aberrant Wnt signaling has been implicated in many cancers, such as colorectal cancer, hepatocellular carcinoma, melanoma, and others [97].

The expression of NDRG1 has been demonstrated to inhibit β-catenin phosphorylation at Ser33/37 and Thr41 [98]. This effect increases the non-phosphorylated β-catenin levels and prevents its nuclear translocation by decreasing the expression of p21-activated kinase 4 (PAK4), which acts as a transporter of β-catenin to the nucleus [98]. This reduction in PAK4 expression prevents β-catenin from acting as a pro-oncogenic transcription factor [82,98]. Moreover, NDRG1 actively promotes β-catenin localization to the plasma membrane, where it acts as a component of the adherens junction, together with E-cadherin, to enhance cell-to-cell adhesion and inhibit tumor cell metastasis [82,98] (Figure 10). Iron-binding ligands, such as DFO and Dp44mT, which up-regulate NDRG1, have been demonstrated to promote the anti-metastatic effects of β-catenin, while inhibiting its oncogenic roles [82].

### 7.3. NDRG1 Inhibits the NF-κB Pathway

Through studies examining PC, it was demonstrated that NDRG1 inhibited the NF-κB pathway by increasing the proteasomal degradation of IκB kinase-β (IKKβ), resulting in decreased IκBα phosphorylation [15,94]. Therefore, a decrease in IκBα phosphorylation results in an increase in the repression of the nuclear translocation of p65 and p50, and the suppressed activation of NF-κB-responsive genes, as more IκBα is available to inhibit the NF-κB dimers [94]. As alluded to previously in Section 2.1, NF-κB has a significant role in promoting metastasis [14]. Hence, the ability of NDRG1 to inhibit the NF-κB pathway is another significant aspect of its anti-metastatic activity.

### 7.4. NDRG1 Suppression of the PI3K/AKT Pathway

As discussed above (Section 1.2), the initial step in the PI3K/AKT pathway is the activation of the PI3K enzyme, which phosphorylates PIP2 to PIP3, which subsequently activates AKT via phosphorylation, forming pAKT [6]. A negative regulator of this pathway is PTEN, which can dephosphorylate PIP3 to PIP2, thereby inhibiting the PI3K/AKT pathway [18]. There is evidence that PTEN and NDRG1 positively regulate each other [18,70], and thus, when NDRG1 expression increases, so too do the PTEN levels. This response is likely the mechanism by which NDRG1 inhibits the PI3K/AKT pathway [18]. In fact, NDRG1 up-regulation decreases the levels of pAKT (Ser^473^) and its downstream target mTOR, while increasing PTEN expression in prostate cancer and PC [18,88]. Therefore, NDRG1 inhibits the pro-oncogenic and survival activity of the PI3K/AKT pathway in cancer, suppressing tumor growth and metastasis.

In addition to the mechanism described above, NDRG1 significantly decreases the expression of the PI3K regulatory subunits phosphor-p85α and phosphor-p55γ, potentially by increasing PTEN expression [88]. Considering this latter suggestion, the phosphorylation of these subunits is crucial for the assembly and activation of the PI3K complex [88,100]. Hence, when the regulatory subunits are de-phosphorylated, the PI3K complex cannot form, which inhibits PI3K signaling [88].

### 7.5. NDRG1 Suppression of the RAS/RAF/MEK/ERK Pathway

NDRG1 inhibits the RAS/RAF/MEK/ERK pathway by up-regulating the expression of the tumor suppressor, SMAD4 [18,88]. SMAD4 subsequently inhibits ERK phosphorylation, thereby blocking the signaling and activity of this pathway [18,88]. The NDRG1-induced increase of SMAD4 expression may also activate its other tumor suppressor effects, such as arresting the cell cycle at G_1_ (acting as a G_1_/S checkpoint) and inducing the expression of the cyclin-dependent kinase inhibitor, p21 [12,22]. However, later-stage cancers can become desensitized to the anti-cancer effects of SMAD4 and the SMAD2/3/4 complex [12]. Under these conditions, the TGF-β/SMAD4 pathway can result in the induction of the EMT [12,22].

### 7.6. NDRG1 Down-Regulates the ERBB Family of Receptors

The expression of NDRG1 has been demonstrated to down-regulate the ERBB family receptor, EGFR [23]. The mechanism of this effect involves up-regulating the tumor suppressor protein, mitogen-inducible gene 6 (MIG6) [101]. Interestingly, NDRG1 does not increase MIG6 levels by acting as a transcription factor, but instead binds to and stabilizes MIG6, extending its half-life [101]. MIG6 binds to EGFR and induces its internalization into early endosomes, which are then trafficked by PTEN to lysosomes for degradation [101]. MIG6 also blocks EGFR activation by inhibiting EGFR dimer formation by direct interaction [101]. Therefore, when NDRG1 stabilizes MIG6 and increases its cellular levels, this increases the access of MIG6 to EGFR, enhancing its down-regulation [101]. It has also been demonstrated that iron-binding ligands such as DFO, Dp44mT and DpC can up-regulate the HIF-1 targets, *MIG6* and *NDRG1*, due to the ability of these agents to increase HIF-1α expression [101]. This effect could enhance the transcription of *MIG6* and *NDRG1*, resulting in their increased mRNA levels, which would theoretically supplement the ability of NDRG1 in stabilizing MIG6. Hence, these iron-binding drugs demonstrate a double enhancing effect in terms of MIG6 expression [101].

Studies have shown that NDRG1 can also down-regulate the expression of other members of the ERBB receptor family, including HER2 and HER3, preventing the formation of active EGFR/HER2 and HER2/HER3 dimers [23]. The ability to down-regulate ERBB family receptors, such as EGFR and HER2/3, is consistent with the broad activity of NDRG1 in inhibiting the numerous oncogenic signaling pathways regulated by these receptors [13,23].

### 7.7. NDRG1 Suppresses Sonic Hedgehog Signaling to Inhibit Desmoplasia

A key characteristic of PC and a contributor to the tenacity and resistance of this tumor is desmoplasia, that is, the formation of a fibrous “capsule” [3]. This capsule surrounds pancreatic tumor cells, and consists of pancreatic stellate cells (PSCs), a dense fibrous extracellular matrix, and immune cells [3], and abrogates the entrance of chemotherapeutic agents leading to therapeutic resistance [4,102]. The PCs secrete factors such as transforming growth factor beta-1 (TGF-β1), fibroblast growth factor-2 (FGF2), sonic hedgehog (SHH), and platelet-derived growth factor (PDGF), which induce PSC activation and subsequently PC cell proliferation and ECM deposition [102]. The activated PSCs also secrete growth factors, such as hepatocyte growth factor (HGF), insulin growth factor-1 (IGF-1), platelet-derived growth factor (PDGF), and stromal cell-derived factor-1, creating a favorable environment for cancer cell growth [102,103].

Focusing on the SHH pathway, it is known that the mutated and constitutively active KRAS is present in more than 90% of PC, and that it activates NF-ĸB, which directly up-regulates SHH in PC cells (Figure 11) [104]. Once it is secreted from PC cells, SHH acts on PSCs to activate them and stimulate desmoplasia by binding to the patched (PTCH1) receptor [105]. The binding of SHH to PTCH1 leads to the activation of the smoothened (SMO) receptor in PSCs, with subsequent signaling leading to GLI1 activation that promotes PC (Figure 11) [105,106]. GLI1 promotes HGF and IGF-1 transcription in PSCs that is then secreted and activates oncogenic c-MET and IGF-1Rβ signaling in PC cells, respectively [105,107]. These growth factors also stimulate PC cells to continue to secrete SHH, creating a positive feedback loop and bidirectional crosstalk between PSCs and PCs, leading to desmoplasia [4].

In addition to the more favourable properties of DpC compared to Dp44mT described above in Section 5.5, it has been demonstrated recently that through the ability of DpC to up-regulate NDRG1, this agent could potently inhibit the bidirectional oncogenic crosstalk between PSCs and PC cells, which promotes desmoplasia [4]. The up-regulation of NDRG1 by DpC inhibited the SHH secretion from PC cells, preventing the activation of PSCs to secrete HGF and IGF-1, which promote the growth and migration of PC cells (Figure 11) [105,107]. Furthermore, DpC was cytotoxic towards PC cells, but not PSCs, and instead “re-programmed” PSCs into an inactive state, underscoring its exciting selectivity [4].

The fact that NDRG1 inhibits such a variety of distinct oncogenic pathways is a testament to its potency and versatility as a tumor and metastasis suppressor. However, the extensive anti-oncogenic effector activity of NDRG1 means that many of its effects are yet to be elucidated. For example, it has been demonstrated that NDRG1 can inhibit Src-family kinases [108], which have many downstream targets, such as Casitas B-lineage lymphoma (c-Cbl), which regulates many critical protein tyrosine kinases, e.g., EGFR [108]. It can be hypothesized that NDRG1’s inhibitory effect on Src-family kinases may decrease the phosphorylation and subsequent oncogenic activity of c-Cbl. In order to gain a better understanding of c-Cbl’s molecular activity and the possible implications of decreasing its phosphorylation via NDRG1, the structure and function of c-Cbl will be described in detail below.

## 8. Casitas B-Lineage Lymphoma (c-Cbl)

### 8.1. Molecular Structure and Function of c-Cbl

The CBL family, in mammals, consists of c-Cbl, Cbl-b and Cbl-3 proteins [109]. c-Cbl is encoded by the *Cbl* gene located on chromosome 11, position 11q23.3, which—when translated—results in a 906 amino acid protein with an estimated molecular weight of 120 kDa [110,111,112]. c-Cbl is primarily a cytoplasmic protein expressed in virtually all cell types. However, it associates with the membrane upon receptor tyrosine kinase (RTK) stimulation/phosphorylation, and has even been shown to localize to immunological synapses in T-cells [109]. Cbl proteins are important E3 ubiquitin ligases that initiate the degradation of proteins by recruiting E2 (ubiquitin-conjugating enzyme), which catalyzes the transfer of ubiquitin (Ub) from E2 to the protein target [109,111,113]. Ubiquitinated proteins are then targeted to the lysosome or proteasome for degradation depending on the pattern of Ub addition [109].

The CBL family regulates immune system components, protein tyrosine kinase signaling, and an array of downstream signaling proteins, primarily by ubiquitination [109,111,113]. All Cbl proteins have an *N*-terminal tyrosine kinase binding (TKB) domain that recognizes specific phospho-tyrosine residues on target proteins, such as RTKs (Figure 12). This TKB domain is followed by a linker helical region (LHR), which also recognizes specific target proteins for ubiquitin conjugation. A RING finger (RF) domain comes after the LHR that recruits E2 and mediates the transfer of Ub to the target protein (Figure 12). A *C*-terminal proline-rich region follows the RF domain and allows interaction with proteins containing the Src homology 3 (SH3) domain (e.g., Grb2). This latter region is followed by a tyrosine-rich sequence, which becomes a binding motif for SH2 (Src homology 2) domain-containing proteins, such as Crk and p85, after phosphorylation [109,111,112,113]. CBL proteins terminate with a ubiquitin-associated domain (UAD) that is crucial for CBL proteins to form dimers [109,111,112,113] (Figure 12).

### 8.2. c-CBL Down-Regulates RTKs

c-CBL is involved in the negative regulation of RTKs such as PDGF, EGFR, or c-Met, acting as a negative feedback mechanism [109,111,113,114]. Growth factor binding induce3s RTK auto-phosphorylation and recruits c-CBL to the activated receptor via the Grb2 adaptor protein. The RTK activates many proteins, including Src kinases that phosphorylate c-Cbl [109] (Figure 13). Src kinases also phosphorylate other proteins, such as Sprouty2 (hSpry2), which associates with c-Cbl in the cytoplasm to inhibit its E3 ligase activity [109]. However, once hSpry2 is phosphorylated, it detaches from the RF domain and interacts with the TKB domain of c-Cbl [109]. This event allows the RF domain to recruit the E2-conjugating enzyme, which induces the polyubiquitination of hSpry2, leading to its degradation [109]. The TKB domain is then free to bind the target receptor that enables c-Cbl to catalyze the addition of Ub to the RTK, resulting in receptor internalization through endocytosis, in turn resulting in the formation of endosomes [109,111,115] (Figure 13). Subsequently, Ub continues to be added and marks the RTK for degradation through the lysosome [109].

In the context of cancer, c-CBL can down-regulate EGFR and c-Met receptors, which play crucial roles in oncogenesis [109,114]. Thus, c-Cbl can act as a tumor suppressor [109,114].

### 8.3. The Regulation of c-Cbl

c-Cbl is phosphorylated on tyrosine residues, the most well-characterized being Tyr371, Tyr700, Tyr731 and Tyr774 [109,113] (Figure 12). These tyrosine residues elicit different functions of c-Cbl upon phosphorylation by Syk- and Src-family kinases [108,109]. For example, the phosphorylation of the Tyr371 residue within the LHR is required for the activation of c-Cbl’s ubiquitination activity, which induces the degradation of protein tyrosine kinases [113,116]. The phosphorylation of Tyr700, Tyr731 and Tyr774 allows c-Cbl to bind several adapter proteins [109], which will be discussed in more detail below.

Not only RTKs can activate c-CBL. Many other stimuli can induce c-Cbl phosphorylation and its subsequent activity, including integrins, prolactin, the ligation of T-cell and B-cell receptors, Fc-receptors, receptors for erythropoietin, GM-CSF, interleukin-3, and thrombopoietin [110]. As described above, c-Cbl is also negatively regulated by a number of factors, such as hSpry2, which binds the RF domain and inhibits Cbl E3 ligase activity [109]. Following RTK stimulation, hSpry2 is then phosphorylated, which leads to its degradation, relieving the inhibition of c-Cbl [109]. The Rho family GTPase, Cdc42, sequesters c-Cbl with the aid of p85Cool/b-Pix and prevents it from interacting with EGFR, inhibiting c-Cbl-mediated degradation [109,117]. In addition, despite functioning as a ligase, c-Cbl itself is negatively regulated by ubiquitylation [109]. Following RTK stimulation, Src is activated and mediates the phosphorylation of c-Cbl on tyrosine residues, including Tyr371, which is speculated to promote the self-ubiquitination and degradation of c-Cbl [109,118]. Other E3 ligases, such as Nedd4, also facilitate the degradation of c-Cbl [119].

### 8.4. c-Cbl and the c-Met/Crk/JNK and PI3K Pathways

The phosphorylation of c-Cbl in the tyrosine-rich *C*-terminal region by Syk and the Src-family kinases allows it to recruit SH2-domain-containing proteins, such as Crk and p85 [109,110]. This recruitment can enhance the signaling pathways of RTKs [109,110].

### 8.5. c-Cbl Enhances the PI3K/AKT Pathway

p85 is a regulatory subunit of the class IA PI3K enzymes, which is the only PI3K class implicated in cancer [6]. The class IA PI3Ks contain a regulatory subunit named p85, with the isoforms p85α, p55α, p50α, p85β and p55γ, and a catalytic subunit, p110, with the isoforms p110α, p110β and p110δ [6,120]. The PI3Ks are activated by RTKs. For instance, following growth factor stimulation, the RTK undergoes auto-phosphorylation, which allows the p85 regulatory subunit to bind to the phosphotyrosine residues of RTKs [6]. This binding prevents p85 from inhibiting the p110 catalytic subunit, and recruits the p85-p110 PI3K heterodimer to the plasma membrane, where its PIP2 substrate is located, initiating the PI3K/AKT signaling pathway [6] (Figure 14). There is also evidence that the p85 regulatory subunit is phosphorylated by RTKs or other kinases, such as Src, that are important in the assembly and activation of the PI3K complex in the PI3K pathway [88,100,121].

When the Tyr731 residue of c-Cbl is phosphorylated, for example after RTK stimulation, this effect enables c-Cbl to bind p85 and recruit PI3K to the RTK at the membrane [109] (Figure 14). This interaction results in enhanced PI3K/AKT pathway activation, as opposed to PI3K directly binding to the RTK via p85, as described above [109]. This enhancement promotes cell survival and proliferation, and is thought to be mediated by the c-Cbl-PI3K interaction [110]. Such a mechanism is supported by studies showing c-Cbl enhancing PI3K activity and the proliferation of cells when exposed to IL-4 and G-CSF [110].

### 8.6. c-Cbl Enhances the c-Met/Crk/JNK Pathway

Considering the c-Met/Crk/JNK pathway, c-Met is activated by the binding of the hepatocyte growth factor (HGF), leading to its autophosphorylation [122]. The adaptor protein, Gab1, recognizes the phosphotyrosine residues on activated c-Met, and binds to the receptor directly or in conjunction with Grb2 [122]. Then, c-Met phosphorylates Gab1, allowing it to recruit Crk to the c-Met receptor, which mediates the phosphorylation of Janus kinase 1 (JNK) via Crk [110,122] (Figure 15). Activated JNK then induces the transformation of cells into a more tumorigenic state [110,122]. Gab1’s recruitment and interaction with Crk has been demonstrated to more effectively enhance JNK activation compared to the Gab1-independent pathways of JNK activation [110].

c-Cbl also becomes phosphorylated upon c-Met activation [110,122,123]. Phosphorylation at Tyr700 and Tyr774 by Src kinases allows c-Cbl to bind to and act as a docking protein for Crk at the c-Met receptor, functioning very similarly to Gab1 [109,110] (Figure 15). Studies have shown that c-Cbl can enhance JNK activation similarly to Gab1, which is likely due to the interaction and recruitment of Crk to the c-Met receptor [110]. Therefore, c-Cbl and Gab1 are utilized in the c-Met/Crk/JNK pathway as docking proteins to enhance downstream signaling [110]. In addition, c-Cbl can down-regulate c-Met and other RTKs through ubiquitination [109,111,122], as described in Section 8.2 (Figure 15). The c-Met RTK is involved in the activation of other oncogenic signaling mechanisms, including the RAS/RAF/ERK and PI3K/AKT pathway, leading to augmented proliferation, cell motility, invasion, and enhanced cellular survival [124].

Overall, c-Cbl promotes PI3K/AKT and c-Met/Crk/JNK signaling pathways via the recruitment of oncogenic proteins, such as Crk and the p85, which facilitate oncogenic signaling [109,110]. The high expression of Cbl is commonly observed in breast cancer cell lines, and also primary breast and prostate tumors [113]. Interestingly, c-Cbl is downregulated in 60% of PDAC cases [125]. Reduced c-Cbl levels disrupt its ability to regulate EGFR activity via lysosomal degradation, leading to increased EGFR dysregulation and aberrant activation in response to stress induced by chemotherapeutics, and may contribute to treatment resistance in PC [125]. However, the PC microenvironment contains abundant growth factors that stimulate the EGFR and c-Met receptors [102]. Hence, it is expected that c-Cbl could facilitate their downstream signaling, as described above, even with decreased expression. This consideration is especially relevant to the remaining 40% of PDAC that do not have decreased c-Cbl expression [125]. Therefore, the pro-oncogenic role of c-Cbl in PC and its response to NDRG1 and NDRG1-inducing therapeutics, such as DpC and Dp44mT, is essential to consider.

## 9. Summary

PC remains one of the most belligerent cancers, with even the most up-to-date chemotherapies proving to be largely ineffective for the improvement of PC survival rates. It is clear that a novel and innovative approach is required for PC treatment, and with the advent of therapeutic compounds of the thiosemicarbazone class, the prospect of effective treatment for PC may be achievable. These latter agents have unique properties allowing the selective targeting and inhibition of cancer cell growth and metastasis through multiple mechanisms, including the depletion of essential metal ions such as iron, which results in the up-regulation of the potent metastasis suppressor, NDRG1.

The NDRG1 protein suppresses a wide variety of oncogenic pathways that contribute to the growth and spread of PC cancer. More recently, NDRG1 has been demonstrated to inhibit the bi-directional crosstalk between PC and pancreatic stellate cells, which is required for desmoplasia formation in pancreatic tumors, one of the key characteristics of PC that makes it resistant to current treatments.

Considering the oncogenic targets suppressed by NDRG1, e.g., c-Src [126], it was hypothesized in this review that NDRG1 could inhibit the ability of c-Src to phosphorylate c-Cbl. Indeed, c-Cbl is a significant E3 ubiquitin ligase involved in the regulation of regulating various signaling pathways via the degradation of protein tyrosine kinases and RTKs such as c-Met and EGFR. NDRG1 may inhibit the phosphorylation of c-Cbl on two key residues (Tyr731 and Tyr774), possibly via its known ability to suppress Src kinase activity. The phosphorylation of Tyr731 and Tyr774 allows c-Cbl to recruit Crk and p85, respectively, and thus the ability of NDRG1 to inhibit the phosphorylation of these sites would prevent c-Cbl from recruiting these oncogenic proteins. Subsequently, this would inhibit the downstream signaling mediated by c-Met/Crk/JNK and the PI3K/AKT pathways, further facilitating NDRG1’s impressive anti-oncogenic effector activity.

## Figures and Tables

**Figure 1 cells-10-02382-f001:**
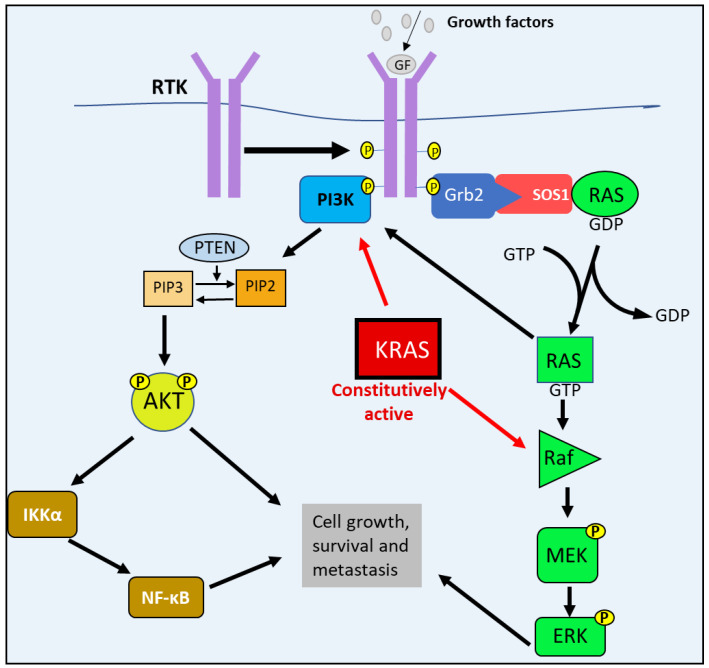
The RAS/RAF/MEK/ERK and PI3K/AKT pathways’ abnormal activation in PC. Following growth factor stimulation, receptor tyrosine kinase (RTK) undergoes auto-phosphorylation, which enables guanine nucleotide exchange factors (GEFs)—such as Son of Sevenless-1 (SOS1)—to bind to the receptor with the aid of adaptor protein GRB2. SOS1 interacts with RAS and catalyzes the exchange of GDP for GTP, forming the active RAS-GTP molecule. Active RAS then associates with and activates RAF, which subsequently phosphorylates MEK that in turn phosphorylates ERK. Ras-GTP can interact with and stimulate PI3K activity. PI3K also binds to active RTKs that initiate the catalytic phosphorylation of PIP2 to PIP3. PIP3 then phosphorylates AKT. The activation of both AKT and ERK leads to the activation of the downstream targets involved in cell growth, survival, and metastasis. Mutations lead to constitutively active KRAS, causing the constant stimulation of PI3K and RAF, leading to PC cell growth, survival and metastasis.

**Figure 2 cells-10-02382-f002:**
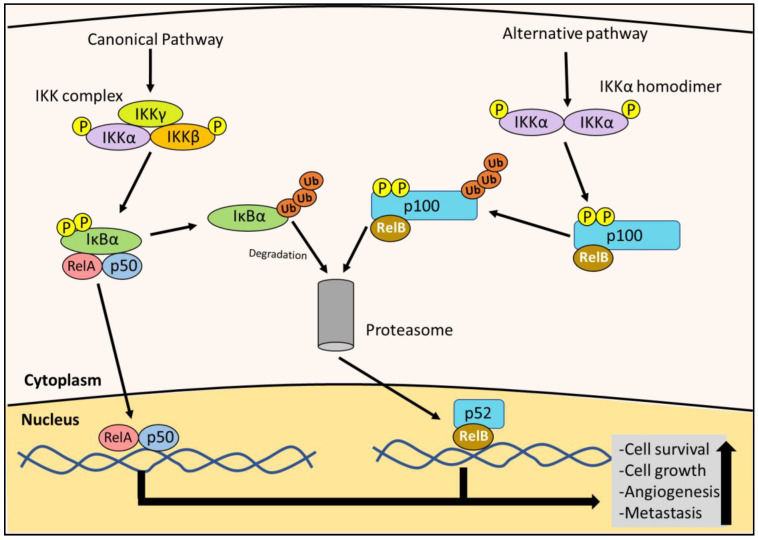
Canonical and alternative NF-κB pathways. As part of the canonical NF-κB pathway, the IκB kinase (IKK) complex is activated by phosphorylation. The IKK complex then phosphorylates the inhibitor of NF-κBα (IκBα), leading to its degradation by the proteasome. This event allows the NF-κB (RelA/c-Rel-p50) dimer to translocate to the nucleus. The alternative pathway is initiated when the IKKα homodimer is activated by phosphorylation, which phosphorylates the p100 bound to RelB. This phosphorylation leads to p100 being degraded by the proteasome to form the RelB-p52 dimer. This NF-κB (RelB-p52) dimer then translocates to the nucleus. Both RelA/c-Rel-p50 and RelB-p52 dimers act as transcription factors to up-regulate the expression of the downstream oncogenic effectors (e.g., c-myc and cyclin D1) that promote cell growth, survival, angiogenesis and metastasis.

**Figure 3 cells-10-02382-f003:**
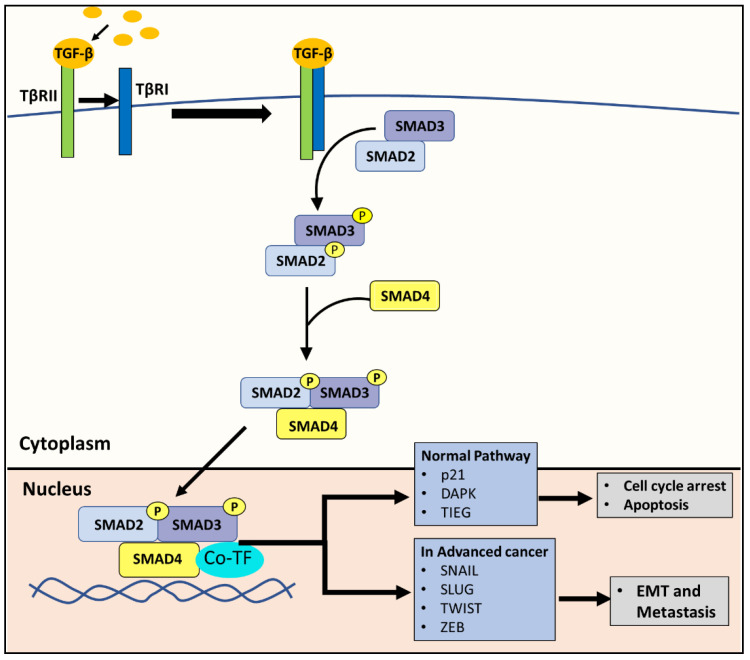
The canonical TGF-β pathway. The TGF-β ligand binds to the TGF-β type II receptor (TβRII). This binding induces TGF-β type I receptor (TβRI) to form a hetero-tetrameric complex with TβRII. TβRI subsequently phosphorylates SMAD2 and SMAD3, which then form a complex with SMAD4. Under physiological conditions in normal cells, the resulting SMAD2/3/4 complex translocates to the nucleus, recruits co-transcriptional factors, and activates genes involved in apoptosis and cell cycle arrest, such as the cyclin-dependent kinase inhibitor, p21, and the death-associated protein kinase (DAPK). In advanced cancer, the same SMAD2/3/4 complex can instead promote the expression of SNAIL, SLUG, TWIST and ZEB, which are transcription factors known to promote the EMT.

**Figure 4 cells-10-02382-f004:**
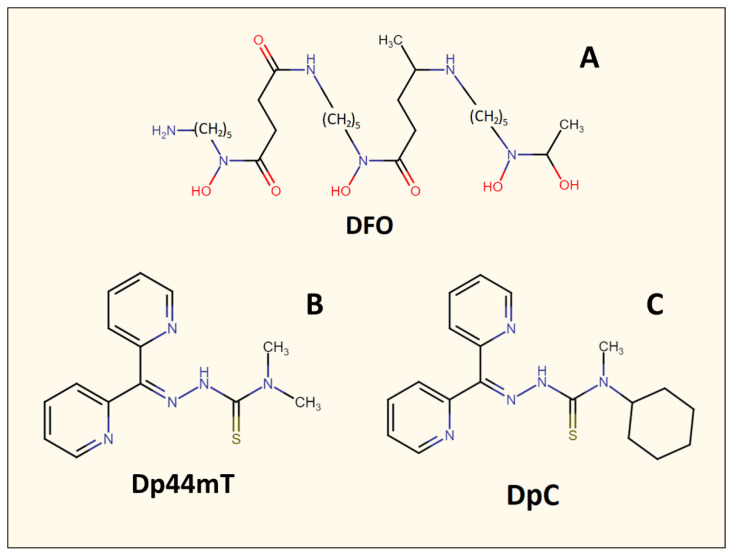
Line drawings of the chemical structures of the metal-binding ligands described herein: (**A**) desferrioxamine (DFO), (**B**) di-2-pyridylketone 4,4-dimethyl-3-thiosemicarbazone (Dp44mT), and (**C**) di-2-pyridylketone 4-cyclohexyl-4-methyl-3-thiosemicarbazone (DpC).

**Figure 5 cells-10-02382-f005:**
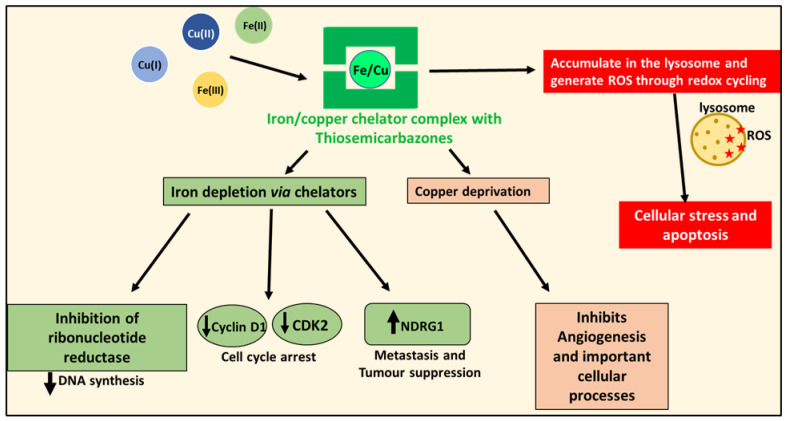
Anti-oncogenic effects of iron and copper-binding ligands. Metal chelators can deplete tumor cell iron and copper. This effect inhibits the iron-dependent enzyme, ribonucleotide reductase, that is required for DNA synthesis, as well as the expression of cyclin D1 and CDK2, resulting in cell cycle arrest. Iron depletion also increases the expression of the potent metastasis suppressor, NDRG1. These complexes also deprive cancer cells of the copper needed for angiogenesis and other crucial cellular processes. Additionally, some ligands—such as Dp44mT and DpC—form iron and copper complexes that accumulate in lysosomes and generate ROS, which leads to lysosomal membrane permeabilization and the apoptosis of cancer cells.

**Figure 6 cells-10-02382-f006:**
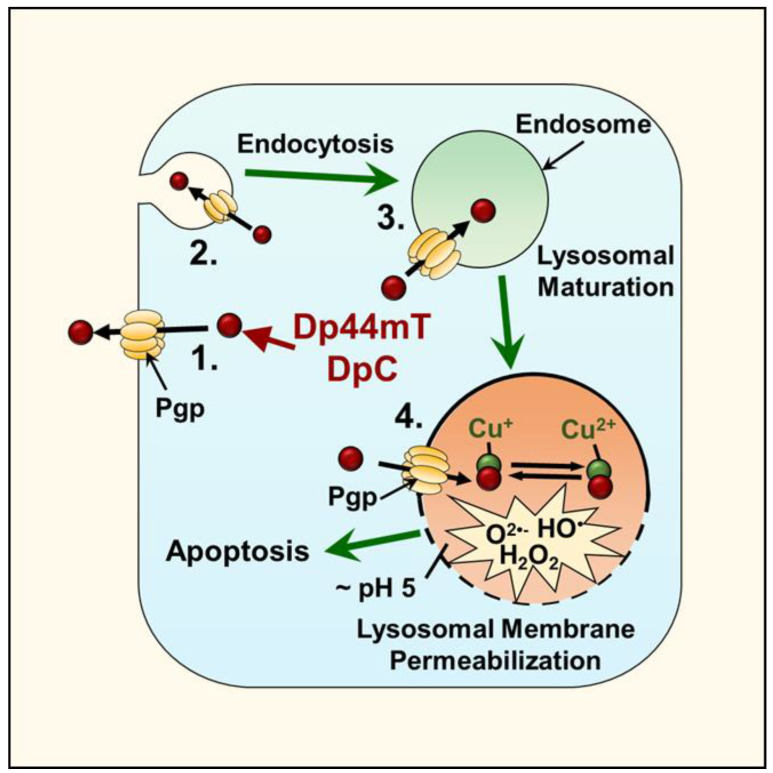
Thiosemicarbazone lysosomal transport and ROS production. Both Dp44mT and DpC use the Pgp transporter to enhance their transport into lysosomes. The expression of Pgp is present on the plasma membrane, and also on endosomal and lysosomal membranes. Plasma membrane endocytosis results in the internalization of Pgp and its presence within endosomes and lysosomes, with this process resulting in the Pgp transporter pumping substrates, such as DpC and Dp44mT, into the lumen of the endosome or lysosome. Upon the entrance of these ligands into the lysosome, their ability to bind the iron and copper derived from the breakdown of intracellular constituents leads to the generation of redox active complexes that induce lysosomal membrane permeabilization and apoptosis.

**Figure 7 cells-10-02382-f007:**
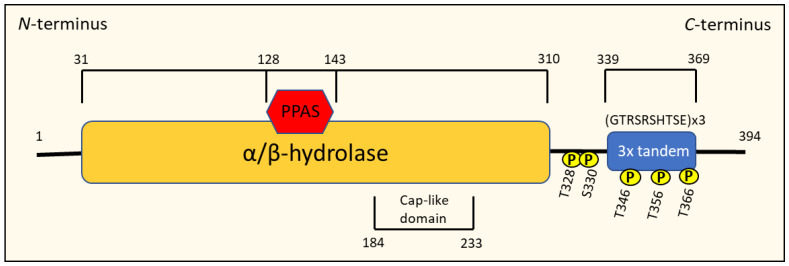
Schematic representation of the structure of the N-myc downstream-regulated gene 1 (NDRG1) protein.

**Figure 8 cells-10-02382-f008:**
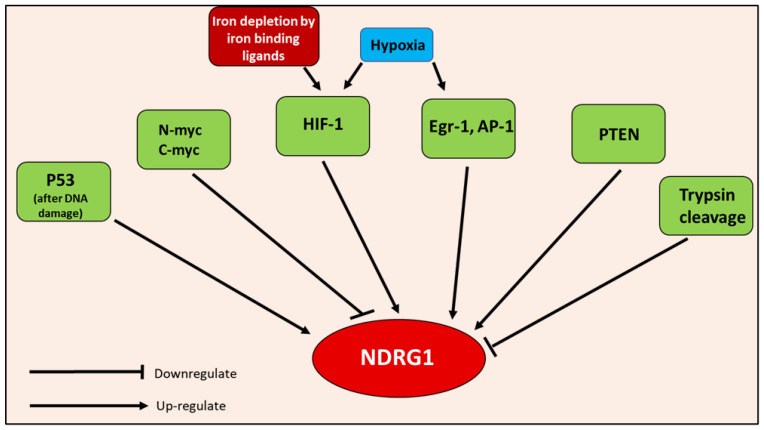
Inducers of NDRG1, including: HIF-1, Egr-1, AP-1, PTEN, p53, hypoxia, iron depletion, and inhibitors of NDRG1, including N-myc, c-myc, and trypsin cleavage.

**Figure 9 cells-10-02382-f009:**
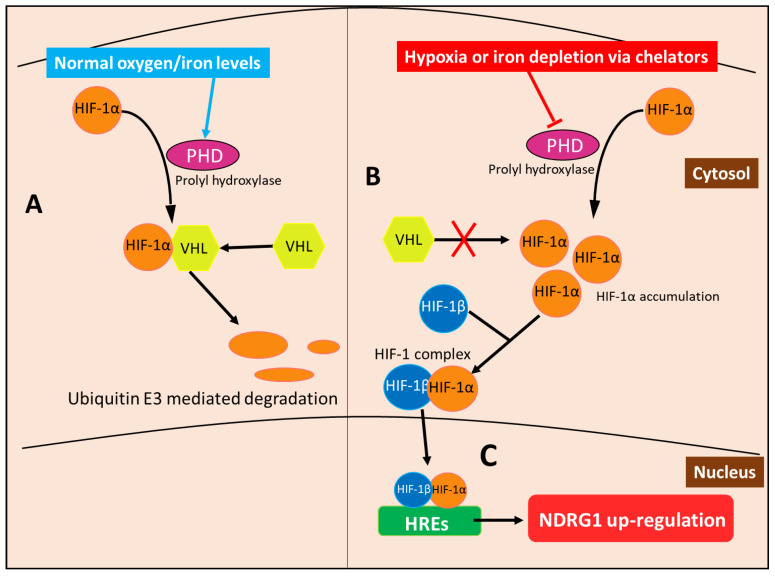
Metal-binding ligands such as DFO induce NDRG1 expression via HIF-1α accumulation. (**A**) Normal oxygen and iron levels enable prolyl hydroxylase (PHD) to hydroxylate HIF-1α, creating a binding site for von Hippel-Lindau (VHL) protein to bind HIF-1α and activate ubiquitin E3 ligase, resulting in the proteasomal degradation of HIF-1α. (**B**) In contrast, during hypoxia or when metal-binding ligands deplete cellular iron, the PHD enzyme is inactivated and does not hydroxylate HIF-1α, preventing the binding of VHL. This mechanism prevents HIF-1α degradation, which consequently accumulates and forms the HIF-1 complex by binding to HIF-1β. (**C**) The HIF-1 complex then translocates to the nucleus and binds the hypoxia-response elements (HREs) located in the *NDRG1* promoter, thus up-regulating its expression.

**Figure 10 cells-10-02382-f010:**
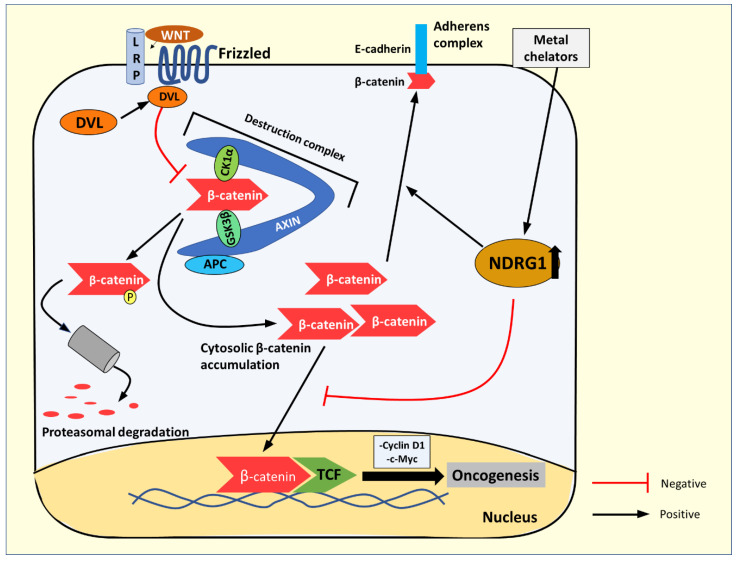
The Wnt/β-catenin signaling pathway and NDRG1’s effect on β-catenin localization. In the absence of the Wnt ligand, β-catenin is phosphorylated by the destruction complex (consisting of casein kinase 1α (CK1α), glycogen synthase kinase 3β (GSK3β), the tumor suppressor APC, scaffold protein AXIN, and others (not shown)) leading to its proteasomal degradation. The Wnt ligand binds to and activates the LRP5/6 and Frizzled co-receptors. The activated receptor then recruits Dishevelled (DVL), where it activates and then sequesters the “destruction complex”, which inhibits its activity, allowing non-phosphorylated β-catenin to accumulate in the cytosol and then translocate to the nucleus. Here β-catenin acts as a co-activator with the T cell factor (TCF) family of transcription factors, which up-regulate proteins such as cyclin D1 and c-myc that promote oncogenesis. NDRG1 up-regulation by iron-binding ligands such as DFO or Dp44mT inhibits β-catenin from translocating to the nucleus, preventing its transactivation activity and promoting cytosolic β-catenin localization to the membrane to form part of the adherens complex.

**Figure 11 cells-10-02382-f011:**
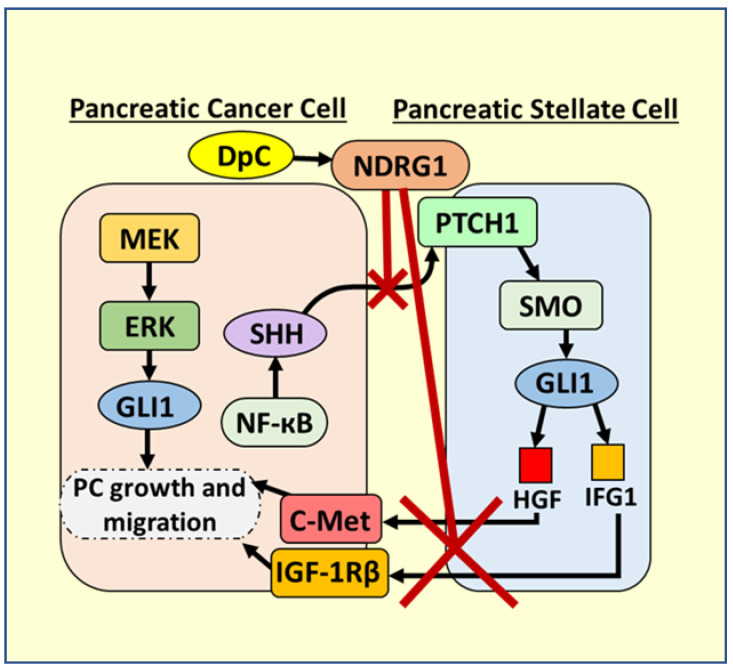
Targeting oncogenic Sonic Hedgehog signaling between PC cells and PSCs via NDRG1 induced by DpC. Pancreatic cancer (PC) cells have an abnormal NF-ĸB activation, which stimulates the production of Sonic Hedgehog (SHH). The secreted SHH binds to the PTCH1 membrane receptor on pancreatic stellate cells (PSCs). The binding of SHH to PTCH1 leads to the activation of the smoothened (SMO) receptor in PSCs that subsequently activates GLI1. GLI1 promotes HGF and IGF-1 transcription in PSCs, which are secreted and bind to c-Met and IGF-1Rβ receptors on PC cells, promoting their growth and migration. HGF and IGF-1 further stimulate SHH production, establishing bidirectional crosstalk between PSCs and PC cells. It was found that the DpC-induced upregulation of NDRG1 potently inhibits this bidirectional crosstalk by inhibiting SHH production by PC cells, preventing the activation of PSCs to secrete HGF and IGF-1.

**Figure 12 cells-10-02382-f012:**
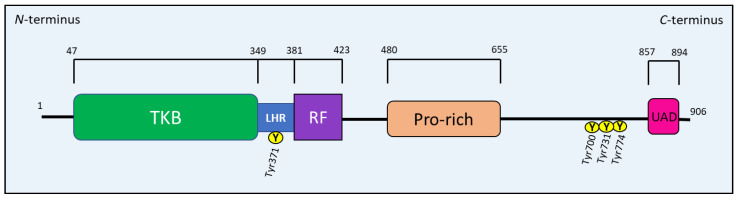
Schematic drawing representing the structure of the Casitas B-lineage lymphoma (c-CBL) protein.

**Figure 13 cells-10-02382-f013:**
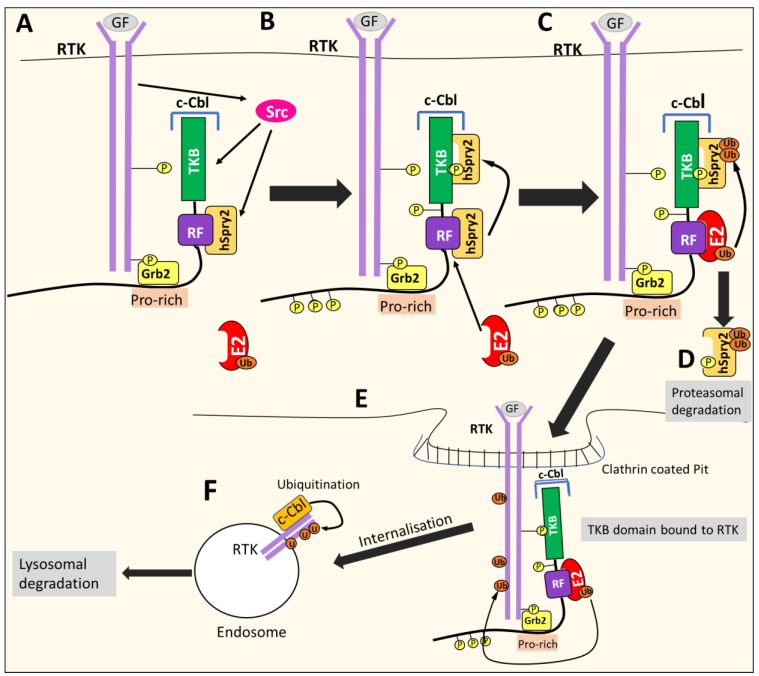
c-Cbl-mediated internalization, ubiquitination and degradation of activated RTKs. (**A**) Growth factor (GF) binding induces RTK auto-phosphorylation and recruits c-CBL to the activated receptor via the Grb2 adaptor protein. The RTK activates Src kinase, which phosphorylates c-Cbl and Sprouty2 (hSpry2), an inhibitor of c-Cbl’s E3 ligase activity. (**B**) Phosphorylated hSpry2 detaches from the ring finger domain (RF) and binds the terminal tyrosine kinase binding (TKB) domain of c-Cbl. (**C**) The RF domain of c-Cbl then recruits the E2-conjugating enzyme, which promotes the polyubiquitylation of hSpry2. (**D**) The polyubiquitylation of hSpry2 targets it for proteasomal degradation. (**E**) The TKB domain of c-Cbl is now free to bind the RTK, which enables c-Cbl to catalyze the addition of ubiquitin (Ub) molecules from E2 to the RTK. The addition of Ub leads to receptor internalization via endocytosis. (**F**) Ubiquitin molecules (Ub) continue to be added, and mark the RTK for lysosomal degradation.

**Figure 14 cells-10-02382-f014:**
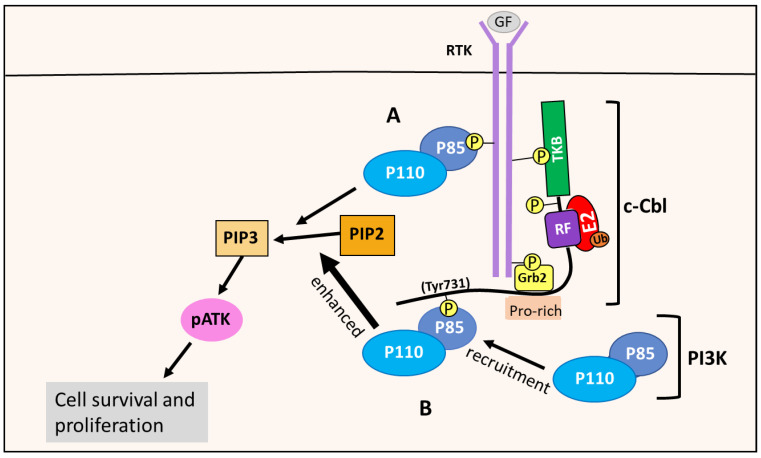
c-Cbl enhances the PI3K/AKT pathway via PI3K recruitment. (**A**) RTK activation allows the p85 regulatory subunit of PI3K to bind to the phosphor-tyrosine residues of RTKs. This interaction relieves the p85 suppression of the p110 catalytic subunit, and allows PI3K to phosphorylate PIP2 to PIP3, initiating the PI3K/AKT pathway that leads to cell survival and proliferation. (**B**) c-Cbl’s Tyr731 residue is phosphorylated after RTK stimulation, enabling it to bind and recruit PI3K via the p85 regulatory subunit. This results in enhanced PIP2 phosphorylation and PI3K/AKT pathway activation, and thuss promotes cellular survival and proliferation.

**Figure 15 cells-10-02382-f015:**
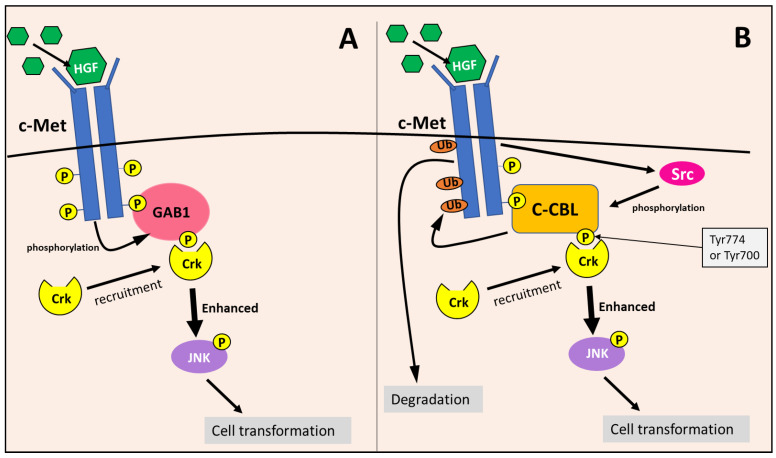
The c-Met/Crk/JNK pathway is enhanced by GAB1 and c-Cbl by promoting JNK phosphorylation. (**A**) c-Met is activated by hepatocyte growth factor (HGF), leading to its autophosphorylation. The adaptor protein, Gab1, recognizes and binds to phosphotyrosine residues on activated c-Met. Then c-Met phosphorylates Gab1 and enables it to recruit Crk (via Tyr residue) to promote JNK phosphorylation. (**B**) Similarly, c-Cbl also recognizes and binds to the phosphotyrosine residues on activated c-Met. Subsequently, c-Cbl is phosphorylated by c-Met-induced Src kinases at either Tyr700 or Tyr774, which enables the recruitment of Crk and the enhancement of JNK phosphorylation. In addition, c-Cbl simultaneously ubiquitinates the c-Met receptor, resulting in its degradation. Both c-Cbl and GAB1 enhance JNK phosphorylation, which promotes cellular transformation.

## Data Availability

No new data were created or analyzed in this study. Data sharing is not applicable to this article.

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
