# Peer review of "The Oncogenic Signaling Disruptor, NDRG1: Molecular and Cellular Mechanisms of Activity"

_cells, 2021, doi:10.3390/cells10092382_

Round 1

Reviewer 1 Report

The extensive review covers the role of NDRG1 protein in oncogenesis on the background of pancreatic cancer. To provide full and comprehensive snapshot of all involved processes, it includes chapters or paragraphs on signal transduction pathways in cancer, NRDG1 connection with iron utilization in cancer cells, and the therapeutic potential of novel iron/copper chelators. Although very complex, the review is clearly written, comprehensible and educative. Numerous diagrammatic figures assist the reading.

Several minor changes can be recommended:

Chapter 3. 

The distinction between adjuvant and neo-adjuvant therapy may not be known to the readers, clarification in the text may assist complete understanding

The manuscript is obviously intended for a wide spectrum of readers, not just for clinicians. Therefore, in addition (or instead) to providing particular drug names (oxaliplatin, irinotecan, nab-paclitaxel, gemcitabine, leucovorin….) readers may appreciate knowing the general category of the drug or a mechanisms of the molecular effect (i.e. DNA-modifiing platinum-based drug, topoisomerase inhibitor, mitotic drug, antimetabolites and folates….).

FOLFIRINOX is misspelled repeatedly in the text as FOLFIRONOX.

FOLFIRINOX includes five drugs – fluoruracil is missing among those mentioned in the text (217-218)

Chapter 6.3

Figure 8 „Chemical and biological agents“ is a very general category. Unless specified, it may cause both stimulation and inhibition of NDRG1. Can be excluded from the figure.

Author Response

Please see the attachment for our comprehensive revision

Reviewer 2 Report

This is a comprehensive review on pancreatic cancer molecular biology, an important and potent mestasis suppressor NDRG1, and iron chelation as relatively novel cancer treatment modality.

The paper nicely connects these individial topics.

The review is well written, I could not find even minor errors or omissions. Also the graphics (15 comprehensive Figures!) is perfect for readers' understanding.

I recommend to publish the paper as it is.

Author Response

Please see the attachment for our comprehensive revision letter

Reviewer 3 Report

In a very detailed review, Chekmarev and colleagues present the role of NDRG1 in tumorigenesis, especially of pancreatic cancer. In addition, the review gives an interesting overview of new therapeutic strategies for the treatment of pancreatic cancer in relation to NDRG1.

However, I have a few notes that the authors should address:

  1. Page 2, lines 67-71: I would strongly recommend to use the term pancreatic neuroendocrine tumor (pNET) instead of endocrine tumor. According tot he latest WHO classification, this is the pathological and clinical nomenclature for these tumors. In addition, contrary to what the authors write 60-90 % of all pNET are non-functional and therefore are asymptomatic and not always readily diagnosable. In contrast, these tumors are often incidentally found. Thus, the authors should revise this passage in the manuscript.
  2. In the entire manuscript, authors should ensure that protein names and genes are spelled consistently (i.e. sometimes the authors write Ras and then again RAS). When talking about genes the authors should uniformly use italic.
  3. Page 6, line 201-202: either enumerate tumors of the colon as follwos: colon, rectum, or summarize as colorectal and not as colon, colorectal, this is just redundant.
  4. Page 6, line 214: Recently, the recommendation for chemotherapy of PDAC was revised. Accordingly, the American Society of Clinical Oncology (ASCO) clinical practice guideline recommends the combination chemotherapy regimen of 5-fluorouracil, oxaliplatin, and irinotecan (modified FOLFIRINOX, also known as mFOLFIRINOX) as the preferred adjuvant therapy for patients with pancreatic cancer and ECOG performance status 0-1who have undergone an R0 or R1 resection and have not received neoadjuvant chemotherapy. For patients with a poor general condition or contraindications to mFOLFIRINOX, gemcitabine monotherapy is available, in the case of intolerance to gemcitabine 5-FU / folinic acid is available as a therapeutic alternative. Thus, the authors have to revise the section about chemotherapeutic regimens in pancreatic cancer. Of note, it is not written FOLFIRONOX but FOLFIRINOX.
  5. Pay attention to the formatting of all headings. In some places you use a full stop at the end of the heading, but in others you don't (i.e. 5.1. without, 5.3. with, etc).
  6. Page 9, line 326: Write glutathione (GSH)
  7. Page 9, line 336 and Page 10, line 343: I guess it should be referred to Figue 6 and not to Figure 7
  8. Page 16, line 564: A bracket is missing behind (GSK3ß
  9. Page 22, line 770: remove space between interleukin and -3
  10. Page 22, line 785: Check for format of heading 8.5.
  11. Page 25, line 865: After the abbreviation for pancreatic cancer as PC, the authors should use the abbreviation consistently throughout the manuscript.
  12. Full stop after Data availability statement

Author Response

Please see our comprehensive response letter attached

Round 2

Reviewer 3 Report

The authors have implemented all suggestions and revised the manuscript.
In the present form the manuscript is very nice and acceptable. Congrats!